# New Chalcone Derivatives Containing 2,4-Dichlorobenzenesulfonamide Moiety with Anticancer and Antioxidant Properties

**DOI:** 10.3390/ijms25010274

**Published:** 2023-12-24

**Authors:** Anita Bułakowska, Jarosław Sławiński, Anna Hering, Magdalena Gucwa, J. Renata Ochocka, Rafał Hałasa, Łukasz Balewski, Justyna Stefanowicz-Hajduk

**Affiliations:** 1Department of Organic Chemistry, Medical University of Gdańsk, Aleja Generała Józefa Hallera 107, 80-416 Gdańsk, Poland; jaroslaw@gumed.edu.pl; 2Department of Biology and Pharmaceutical Botany, Medical University of Gdańsk, Aleja Generała Józefa Hallera 107, 80-416 Gdańsk, Poland; anna.hering@gumed.edu.pl (A.H.); magdalena.gucwa@gumed.edu.pl (M.G.); jadwiga.ochocka@gumed.edu.pl (J.R.O.); 3Department of Pharmaceutical Microbiology, Medical University of Gdańsk, Aleja Generała Józefa Hallera 107, 80-416 Gdańsk, Poland; rafal.halasa@gumed.edu.pl; 4Department of Chemical Technology of Drugs, Medical University of Gdańsk, Aleja Generała Józefa Hallera 107, 80-416 Gdańsk, Poland; lukasz.balewski@gumed.edu.pl

**Keywords:** chalcone, benzenesulfonamide, synthesis, antitumor, antioxidant, antimicrobial activity, neutrophil elastase inhibition

## Abstract

Chalcones and their derivatives, both natural and synthetic, exhibit diverse biological activities. In this study, we focused on designing and synthesizing (*E*)-2,4-dichloro-*N*-(4-cinnamoylphenyl)-5-methylbenzenesulfonamides **4**–**8** with the following two pharmacophore groups: 2,4-dichlorobenzenesulfonamide and chalcone. The obtained compounds displayed notable anticancer effects on various human cancer cells, such as cervical HeLa, acute promyelocytic leukemia HL-60, and gastric adenocarcinoma AGS, when assessed with the MTT test. The activity of all compounds against cancer cells was significant, and the obtained IC_50_ values were in the range of 0.89–9.63 µg/mL. Among all the tested compounds, derivative **5** showed the highest activity on the AGS cell line. Therefore, it was tested for cell cycle inhibition, induction of mitochondrial membrane depolarization, and activation of caspase-8 and -9. These results showed that this compound strongly arrested the cell cycle in the subG0 phase, depolarized the mitochondrial membrane, and activated caspase-8 and -9. Similar to the anticancer effects, all the obtained compounds **4–8** were also assessed for their antioxidant activity. The highest antiradical effect was demonstrated for derivative **5**, which was able to inhibit DPPH and ABTS radicals. All examined compounds showed dose-dependent activity against neutrophil elastase. Notably, derivatives **7** and **8** demonstrated inhibitory properties similar to oleanolic acid, with IC_50_ values of 25.61 ± 0.58 and 25.73 ± 0.39 µg/mL, respectively. To determine the antibacterial activity of derivatives **4**–**8**, the minimum bacteriostatic concentration (MIC) values were estimated (>500 µg/mL for all the tested bacterial strains). The findings demonstrate the substantial potential of sulfonamide-based chalcone **5** as a promising drug in anticancer therapy.

## 1. Introduction

The quest for novel medicinal substances has been a companion to humans throughout history. Before the mid-19th century, natural resources were utilized to treat many ailments. The first successful attempts to isolate biologically active substances from plants paved the way for the development of organic chemistry and the discovery of the correlation between the structures of chemical compounds and their biological activities. These findings led to the introduction of numerous synthetic drugs into medicine. Presently, almost a quarter of medicinal substances in everyday clinical practice contain active ingredients derived from plant-based origin. Chalcones, categorized as unsaturated aromatic ketones, naturally occur in many plants. In pharmacology, they belong to the group of polyphenolic compounds, and their activity is attributed to the presence of a conjugated unsaturated bond to the carbonyl group (1,3-diaryl-2-propen-1-one) [1,2,3]. They exist as aglycones or in glycosidic and ester-linked forms in nature. So far, their presence has been confirmed in a few species of plants, such as licorice (*Glycyrrhiza glabra*, Fabaceae—isoliquiritigenin and its glycoside isoliquirititoside), hops (*Humulus lupulus*, Cannabinaceae—xanthohumol), immortelle (*Helichrysum arenarium*, Asteraceae), and willows (*Salix* sp., Salicaceae—isosalipurposide) [4,5]. Chalcones and their derivatives, whether natural or synthetic, exhibit a diverse array of biological activities, including anticancer [6,7,8], antibacterial [9], antifungal [10,11], antiviral [12], antiprotozoal [13], antioxidant [14,15], and anti-inflammatory [16] properties. Their increasing popularity in the field is due to their strong antioxidant properties and anticancer effects [17]. The adaptable chemical skeleton of 1,3-diaryl-2-propen-1-one can be conveniently modified to change the biological activity of the molecule by adding various functional groups [18]. This versatility enables chalcone to bind to various molecular targets, making them a useful group for the development of novel anticancer agents. The integration of chalcone moiety with other anticancer pharmacophores to create hybrid compounds has the potential to overcome drug resistance and is a promising strategy for the development of new drugs with therapeutic specificity [19,20,21,22]. On the other hand, sulfonamides are organic compounds, recognized as the first synthetic chemotherapy drugs with the SO_2_NH_2_ group. Currently, they have numerous applications in medicine and demonstrate a wide spectrum of biological activity. The anticancer properties of sulfonamides result from different mechanisms of action. Indisulam, for instance, disrupts the cell cycle in the G1 phase and serves as a carbonic anhydrase inhibitor. Preclinical tests on colon cancer cells (HTC-116) showed a high activity of this compound with an IC_50_ of 0.11 µg/mL, and it is currently undergoing the second phase of clinical trials [23]. Another example is vemurafenib (PLX4032 and RG7204), a selective BRAF serine–threonine kinase inhibitor inducing autophagy, which is employed in the treatment of advanced melanoma with a BRAF gene mutation (IC_50_ = 31 nM) [24].

Reports in the literature confirm that chalcones in combination with sulfonamides exhibit a wide spectrum of biological activities [25,26], including anticancer, enzyme inhibition, antioxidant, antidiabetic, and antimicrobial. Sulfonamide moiety has been shown to engage in electrostatic and noncovalent bonding interactions with protein residues at the receptor binding site, which provides sulfonamide-based compounds with the ability to inhibit various enzymes, such as tyrosinase [27], α-glucosidase [28,29], α-amylase [28,29], β-amylase [29], β-secretase, and acetylcholinesterase [30]. Genotoxic effects of sulfonamidochalcones have also been proven [31] and are connected with the inhibition of tubulin formation through the hydrophobic interaction between the oxygen atom of the sulfonamide moiety and the amine hydrogen present in the lysine residues of tubulin. The anticancer effects of such compounds have been evaluated in vitro on several cancer cell lines, including human liver cancer (HEPG2) [32], breast cancer (MCF-7) [33], glioblastoma multiforme (SF-295), prostate cancer (PC-3), and colorectal cancer (HCT-116) [34]. The in vitro anti-inflammatory and antioxidant potential of *p*-sulfonamidochalcones has also been confirmed. The activity was attributed to the presence of the phenyl group of the sulfonamide moiety, which is likely involved in hydrophobic interactions with appropriate residues in the cyclooxygenase-2 (COX-2) binding pocket [35]. In turn, azidosulfonamide chalcone derivatives have been tested for their antimicrobial activity against a wide range of Gram-positive and Gram-negative bacterial strains and fungi [36].

The current study is a continuation of our team’s research on innovative hybrid compounds [37,38]. We designed and synthesized (*E*)-2,4-dichloro-*N*-(4-cinnamoylphenyl)-5-methylbenzenesulfonamide derivatives incorporating the following two pharmacophore groups: 2,4-dichlorobenzenesulfonamide (A) and chalcone (B), as illustrated in Figure 1. The primary objective of our research was to assess the anticancer, antioxidant, and antimicrobial effects of the obtained chalcone–sulfonamide hybrids.

## 2. Results and Discussion

### 2.1. Chemistry

The synthesis of new chalcone derivatives involved a two-stage process. In the initial stage, *N*-(4-acetylphenyl)-2,4-dichloro-5-methylbenzenesulfonamide (**3**) was crucially obtained as a substrate for derivative production. This was achieved by reacting 4-aminoacetophenone (**1**) with 2,4-dichloro-5-methylbenzenesulfonyl chloride (**2**) in dry pyridine (Figure 1) at boiling temperature for 4 h, resulting in 95% yield. Subsequently, the obtained acetophenone underwent Claisen–Schmidt condensation with suitable benzaldehyde derivatives in a basic medium. This synthesis was carried out at room temperature (20–22 °C) for varying periods (24–48 h), yielding five derivatives of (*E*)-2,4-dichloro-*N*-(4-cinnamoylphenyl)-5-methylbenzenesulfonamide **4**–**8** with 53–87% yield.

To confirm the structure of the obtained compounds (**4**–**8**), spectroscopic methods (IR, ^1^H NMR, and ^13^C NMR) and elemental analysis (C, H, and N) were used, with the results detailed in the experimental section. The progression of the reaction and homogeneity of the obtained products were confirmed by thin-layer chromatography (TLC).

The IR spectra of the newly synthesized (*E*)-2,4-dichloro-*N*-(4-cinnamoylphenyl)-5-methylbenzenesulfonamide derivatives **4**–**8** displayed absorption bands of NH groups in the range of 3190–3164 cm^−1^. Additionally, in the fingerprint region, asymmetric and symmetric vibrations of the sulfonyl group were observed in the ranges of 1355–1351 and 1163–1160 cm^−1^, respectively. Moreover, the spectra revealed an absorption band of the carbonyl group in the region of 1652–1646 cm^−1^.

In the analysis of the ^1^H NMR spectra of the series of derivatives **4**–**8** in the aliphatic part, we observed a signal coming from the methyl group in the fifth position of the benzene ring in the range of 2.31–2.49 ppm. Aromatic proton signals ranged from 6.79 to 8.06 ppm. Two doublets originating from alkenyl protons were visible in the range of 7.64–7.82 ppm with coupling constants of 15.6–16.00 Hz, indicating a trans isomer. Two sharp singlets originating from the H-3 and H-6 protons of the benzene ring occurred at 7.48–7.81 and 7.95–8.18 ppm, respectively. Protons of NHSO_2_ groups were visible in the form of a singlet within the range of 11.27–11.33 ppm.

MS spectra were performed for all compounds and are presented in the Appendix A.

### 2.2. Biological Studies

#### 2.2.1. Anticancer Activity

##### Cytotoxic Activity of the Chalcone Derivative Compounds on Cancer Cells

To assess the cytotoxic activity of the five tested compounds on cancer cells and fibroblasts, an MTT assay was used. The activity of all the compounds toward cancer cells was notably significant with IC_50_ values in the range of 0.89–9.63 µg/mL. Among these chalcone derivatives, compounds **5** and **7** exhibited the highest activity with IC_50_ values below 1.0 and 1.57 µg/mL for AGS and HL-60 cells, respectively. For the HeLa cell line, these values were 5.67 ± 0.35 and 6.34 ± 0.04 µg/mL, respectively. On the other hand, compounds with hydrogen, fluorine, and methyl group substituents in the R position (compounds **4**, **6,** and **8**) demonstrated weaker activity (Table 1).

Simultaneously, the tested compounds demonstrated a negligible impact on noncancerous cells (human neonatal fibroblasts) with IC_50_ values ranging from 25.59 to 32.15 µg/mL. These effects were notably 30 times lower than the activities observed for the most potent compounds on cancer cells. Also, within this study, we calculated selectivity indexes for the tested compounds using a specific Formula (1):SI = IC_50_ noncancer cells/IC_50_ cancer cells(1)

A compound was considered selective for cancer cells when SI ≥ 10.

The best SI value was obtained for compound **5** (Table 2). This chalcone derivative was chosen for further experiments determining the cell cycle inhibition and the type of cell death induced in gastric cancer cells.

##### Compound **5** Induced Cell Death in AGS Cells

In the subsequent step of the anticancer study, microscopic observation confirmed that derivative **5** had a strong cytotoxic effect on gastric cancer cells, inducing morphological changes, including cell shrinkage, membrane blebbing, and cell detachment. These changes were clearly visible when higher concentrations of the compound were used in the study. Similar changes were observed in the cells treated with a positive control, oxaliplatin (Figure 2).

In another separate experiment, annexin and 7-aminoactinomycin (7-AAD) were utilized to estimate the amount of live, early, and late apoptotic and dead cells after the treatment of AGS cells with compound **5**. These results indicated that this compound significantly triggered cell death, especially at concentrations above 5 µg/mL. The percentages of late apoptotic and dead cells were 33.92% ± 3.05% and 18.28% ± 1.80%, respectively, for the compound concentration of 5 µg/mL. At the highest concentration used (10 µg/mL), the amount of late apoptotic and dead cells increased to 49.90% ± 2.07% and 22.16% ± 2.04%, respectively. The viability of the cells decreased from 91.29% ± 0.28% to 26.26% ± 4.03% after treatment with the compound (Figure 3).

##### Compound **5** Triggered Depolarization of Mitochondrial Membrane in AGS Cells

To confirm the cytotoxic effect of compound **5** on AGS cells, we determined the changes in cellular mitochondrial potential, considering them one of the indicators of cellular stress and apoptosis. In the experiment, the cells were treated with compound **5** for 5 h and the MitoPotential dye (a lipophilic dye) to detect potential changes, along with 7-aminoactinomycin as a cell-death indicator. The MitoPotential dye accumulates within the inner intact mitochondrial membrane and gives high fluorescence. A decreased fluorescence is observed in cells with depolarized membranes of mitochondria. The obtained results showed a significant decrease in the potential of mitochondrial membrane in AGS cells upon treatment with compound **5**. The amount of depolarized/live cells increased from 4.71% ± 0.55% for the control cells incubated with DMSO (0.2%, *v*/*v*) to 34.20% ± 2.57% for the compound concentration of 10 µg/mL. Simultaneously, the percentages of depolarized/dead cells was 5.53% ± 1.04% for the cells treated with DMSO (0.2%, *v*/*v*) and 32.71% ± 3.50% for the highest used compound concentration (Figure 4). Notably, for oxaliplatin (used at a concentration of 40 µg/mL), no observable changes were seen in the percentages of live, depolarized, and dead cells in comparison to the control after 5 h of treatment.

##### Compound **5** Arrested Cell Cycle in subG0 Phase in AGS Cells

To determine cell cycle arrest, AGS cells were exposed to chalcone derivative **5** at concentrations of 1, 5, and 10 µg/mL for 48 h. The obtained results indicated that this compound induced cell cycle arrest in the subG0 phase. The amount of cells in subG0 was 2.05% ± 0.35% for the control cells treated with DMSO (0.2%, *v*/*v*) and increased to 27.1% ± 0.87% for the compound concentration of 10 µg/mL. The percentages of cells in the G1, S, and G2/M phases decreased from 41.25% ± 1.06% to 34.47% ± 2.87%, 15% ± 1.70% to 13.78% ± 1.70%, and 42.85% ± 3.66% to 21.50% ± 1.50% for the control cells treated with DMSO (0.2%, *v*/*v*) and the compound concentration of 10 µg/mL, respectively (Figure 5). In the case of oxaliplatin, we also observed a significant increase in the amount of AGS cells in the subG0 phase.

The robust cell cycle inhibition in the subG0 phase suggested that chalcone derivative **5** might induce apoptosis in gastric cancer cells.

##### Compound **5** Increased the Activity Level of Caspase-8 and -9 in AGS Cells

To estimate the activity level of caspase-8 and -9 in gastric cancer cells treated with compound **5** for 24 h, luminometry was utilized. The assay involved luminogenic caspase-8 and -9 substrates, luciferase activity, and cell lysis. The data indicated that compound **5** significantly increased the activity level of caspase-8 in the treated cells with a relative activity results of 1.16 ± 0.15, 3.53 ± 0.33, 3.39 ± 0.28, and 2.99 ± 0.20 for the compound concentrations of 1, 5, 7, and 10 µg/mL, respectively. Stronger activation was observed for caspase-9, and the results were 0.85 ± 0.03, 4.54 ± 0.20, 4.50 ± 0.14, and 3.81 ± 0.32 for the compound concentrations of 1, 5, 7, and 10 µg/mL, respectively. Similarly, in the case of oxaliplatin, a notable increase in the activity of cellular caspase-8 and -9 was observed (Figure 6). These results indicated that compound **5** triggered apoptosis, with both the extrinsic (death receptors) and intrinsic (mitochondrial) pathways being implicated in this process.

The chalcone derivatives tested in this study demonstrated cytotoxic activities with IC_50_ values ≤9 µg/mL, with the lowest values observed in the case of AGS and HL-60 cells. Among them, compound **5** exhibited the most potent activity in the gastric cancer cell line. Thus, compound **5** was chosen for further investigation to study its impact on the cell cycle inhibition and induction of apoptosis. The results indicated that chalcone derivative **5** strongly depolarized the mitochondrial membrane, arrested the cell cycle in the subG0 phase, and caused cell death. This compound also activated caspase-8 and -9, indicating that both the extrinsic apoptosis pathway connected to membrane death receptors (DRs) and the intrinsic mitochondrial pathway were involved in this process. According to the mechanism of apoptosis induced by death ligands and DRs, a death-inducing signaling complex (DISC) is formed. It consists of the DD-containing Fas-associated death domain (DD) as an adaptor molecule, procaspase-8, procaspase-10, and the cellular FLICE inhibitory proteins (c-FLIPs) [39]. The active caspase-8 can cleave Bid (BH3 interacting domain death agonist), which translocates to mitochondria and induces cytochrome c release. The cytochrome c combines with Apaf–1 and procaspase-9 to form an apoptosome that triggers caspase-9 and executioner caspase activation. In turn, these proteases activate cytoplasmic endonucleases, causing chromatin condensation and the formation of apoptotic bodies. In summary, the extrinsic apoptosis pathway can be enhanced by the intrinsic one, in which changes in mitochondrial membrane permeabilization, loss of MMPs, and activation of caspase-9 are observed [40]. However, a more in-depth study is required to identify the genes at the mRNA level that are upregulated during cell death.

Chalcone derivatives containing a sulfonamide moiety were studied on 60 different human cancer cell lines [41]. In these experiments, 12 synthetic compounds were tested, of which 4 had the most potent cytotoxic activity. Specifically, two compounds with chlorophenylacryloyl moiety attached to methoxybenzenesulfonamide had strong activity on K-562 leukemic, HCT-116 colon, LOX IMVI melanoma, and MCF-7 breast cancer cell lines with GI_50_ values of 0.57, 1.36, 1.28, 1.30 µM and 2.77, 1.56, 1.63, and 1.97 µM, respectively. In comparison to the above compounds, the subsequent chalcone–sulfonamide hybrids—methoxytolylacryloylbenzenesulfonamide and 5-cinnamoyl-2-methoxybenzenesulfonamide—also showed a strong cytotoxic effect on K-562 cells, reducing the viability of leukemia cells to less than 10% at 10 µM [41]. In another study, synthetic chalcone derivatives bearing a benzenesulfonamide group were evaluated in vitro for their antitumor effect on HCT-116 colon, MCF-7 breast, and 143B osteosarcoma cell lines. The IC_50_ values of 9 tested compounds on those cancer cells ranged from 0.60 to 19.99 µM. The most active compound with methylbenzenesulfonamide moiety was (*E*)-*N*-{3-[3-(1*H*-benzo[*d*]imidazol-2-yl)-3-oxoprop-1-en-1-yl]phenyl}-3-methylbenzenesulfonamide and displayed IC_50_ values of 0.60, 0.89, and 0.79 µM on HCT-116, MCF-7, and 143B cell lines, respectively. Moreover, molecular docking studies and enzymatic assays revealed that the antitumor activity of this compound might be regulated by the cysteine cathepsin inhibitors Cat L and Cat K, potentially reducing tumor growth, invasion, and metastasis [42].

The specific molecular mechanisms underlying the impact of chalcone–sulfonamide hybrids on cancer cells have not been elucidated to date. However, numerous studies have detailed the anticancer mechanisms of many individual chalcones and sulfonamides. For instance, cardamonin, a chalcone derived from *Campomanesia adamantium*, decreased NF-κB1 activity and increased DNA fragmentation in human prostatic adenocarcinoma PC-3 cells [43]. In hepatocellular carcinoma HepG2 cells, cardamonin exhibited significant cytotoxicity with an IC_50_ value of 17.1 ± 0.592 μM. This compound also caused cell cycle arrest in the G1 phase and induced apoptosis through the activation of both the extrinsic and intrinsic pathways [44]. This compound also showed the ability to reduce the resistance of cancer cells to therapy, as observed in combination with 5-fluorouracil [45]. In this study, cardamonin notably enhanced the chemosensitivity of gastric cancer cells to 5-fluorouracil by suppressing the Wnt/β-catenin signaling pathway. Furthermore, lonchocarpin, extracted from *Lonchocarpus sericeus*, was tested on colorectal cancer cell lines HCT-116, SW480, and DLD-1, and it inhibited Wnt/β-catenin signaling by impairing β-catenin nuclear localization [46]. Also, human neuroblastoma SK-N-SH cells treated with lonchocarpin induced phosphorylation of the AMP-activated protein kinase [47]. In lung cancer H292 cells, lonchocarpin significantly decreased cell proliferation by modulating the Bax/caspase-9/caspase-3 pathway [48].

Numerous sulfonamides with known cellular mechanisms are used successfully as drugs in anticancer therapy. For example, tasisulam exhibits selective toxicity toward tumor cells and induces apoptosis by a mitochondrial-targeted mechanism involving the loss of MMP, induction of reactive oxygen species (ROS), and caspase-dependent cell death [49,50]. However, another sulfonamide indisulam inhibits cyclin-dependent kinases (CDK), induces G1/S phase cell cycle arrest, and may lead to the induction of apoptosis and the inhibition of tumor cell proliferation [51]. Pazopanib is a multiple protein kinase inhibitor and inhibits tumor growth by targeting angiogenesis through the inhibition of vascular endothelial growth factor receptor VEGFR [52]. Encorafenib, on the other hand, inhibits the activation of the RAF/MEK/ERK signaling pathway and the proliferation of BRAF-V600E-mutated tumor cells [53,54,55].

#### 2.2.2. Antioxidant Activity

DPPH and ABTS assays are simple colorimetric methods that provide repeatability of results despite the sample polarity [56]. The β-carotene bleaching method is used in vitro to determine lipid peroxidation based on the loss of the yellow color of β-carotene due to the reaction with linoleic acid and subsequent oxidation in an emulsion. This rate of β-carotene bleaching can be slowed down in the presence of antioxidants, and according to the literature, chalcones have the potential to reduce lipid peroxidation [57,58].

The results presented in Figure 7 exhibit the modest potential of the analyzed chalcone derivatives in the protection of lipids oxidation compared to the strong antioxidant ascorbic acid (IC_50_ = 15.6 ± 0.65 μg/mL). Results obtained for chalcone derivatives were similar for compounds **5**, **6**, and **7**, whereas chalcones with R = –H or –CH_3_ (compounds **4** and **8**) exhibited weaker properties to reduce lipid peroxidation.

In DPPH and ABTS assays, the results indicated a limited capacity of the tested chalcone derivatives to scavenge free radicals. The widely used reference compound, ascorbic acid, at the concentration of 800 µg/mL exhibited complete inhibition (100%) in both analyses with calculated IC_50_ = 12.75 ± 0.25 and 20.43 ± 0.79 µg/mL in the DPPH and ABTS tests, respectively. However, the chalcones in the amount of 800 µg/mL could not inhibit the radical’s activity by 50% (Table 3). The increasing concentrations of the tested compounds in the reaction buffers resulted in the formation of precipitate. Among the tested compounds, the highest antiradical activity was indicated by compound **5**, which was capable of inhibiting DPPH radical by 30.18% ± 0.23% and ABTS by 16.01% ± 0.69%. Additionally, the other compounds **4**, **6**, **7**, and **8** were unable to react with the ABTS radical.

The obtained data align with previous reports regarding the poor ability of chalcone derivative compounds to scavenge free radicals [58]. According to the literature, utilization of DPPH for the antiradical assessment of chalcone–sulfonamide hybrids presented a range of results depending on the moiety. In particular, better results were obtained for the compound (*E*)-1-(2-aminophenyl)-3-(3,4-dihydroxyphenyl)prop-2-en-1-one in comparison to ascorbic acid. In this study, compounds without the presence of two hydroxyl groups in ring B exhibited no influence on the DPPH radical [59]. The weak antioxidant effect might also be explained by the lack of involvement of the amino group in the compound [59,60].

In our study, the lack of hydroxyl groups that could suppress free radicals may explain the weak antiradical potential of the analyzed chalcone derivatives. In the presented compounds, slight antioxidant properties may result from the presence of an alkenyl moiety. At the same time, the limited antioxidant capacity of this moiety was dispersed as a result of enlarging the molecule with an additional substituent group R in chalcone. As a result, the electron density of the alkenyl moiety decreased.

However, it should be noted that the analyzed compounds have a certain potential to inhibit the oxidation of lipid membranes, and the use of a different model will be considered in further research.

#### 2.2.3. Neutrophil Elastase Inhibition

Neutrophil elastase performs various functions in the body, but its increased activity leads to pathological inflammatory conditions. Inhibition of neutrophil elastase is an emerging trend in the search for compounds that can effectively influence enzyme efficiency to combat lifestyle diseases [61,62].

The impact of chalcone derivatives on the activity of neutrophil elastase was determined with oleanolic acid as a reference (Figure 8). All analyzed compounds exhibited dose-dependent activity on neutrophil elastase. Inhibition properties were detected for compounds **7** and **8** (IC_50_ of 25.61 ± 0.58 and 25.73 ± 0.39 µg/mL, respectively), comparable with oleanolic acid (IC_50_ = 20.45 ± 0.78 µg/mL). Compounds **4**, **5**, and **6** also represented significant antielastase activity but with IC_50_ values above 50 µg/mL (IC_50_ of 52.98 ± 0.24, 63.36 ± 0.84 and 83.83 ± 0.31 µg/mL, respectively), resulting in a better antielastase effect than natural chalcones [63].

#### 2.2.4. Antimicrobial Activity

##### Minimum Inhibitory Concentration Determination

To determine the antibacterial activity of synthetic derivatives, the minimum bacteriostatic concentration (MIC) values were estimated. Across the tested concentration range of compounds, the inhibitory concentration values exceeded 500 µg/mL for all the derivatives and bacterial strains, as shown in Appendix A.

The series of novel chalcone-based sulfone and bisulfone hybrids exhibited MIC values in the range of 1.95–16.62 μg/mL against *B. subtilis* and demonstrated antibacterial activity against *S. typhimurium* (MIC = 1.95 μg/mL). Other hydroxychalcone–sulfonamide derivatives showed strong activity against *B. cereus*, *S. aureus*, and *E. coli* [64]. Additionally, hydroxy- and methoxy-substituted chalcone–sulfonamide hybrids were significantly active on *S. epidermidis* and *P. aeruginosa* with IC_50_ values below 8 µM [65]. In a study by Yamali et al., the replacement of the benzene ring with the indole ring in chalcone–sulfonamide derivatives resulted in much weaker antimicrobial activity against *S. aureus* ATCC 25925, *B. subtilis* ATCC 6633, and *E. coli* ATCC 25923 with MIC values in the range of 125–500 µg/mL [66].

## 3. Materials and Methods

### 3.1. Anticancer Study

#### 3.1.1. Cell Culture

The human gastric adenocarcinoma AGS, cervical cancer HeLa, and acute promyelocytic leukemia HL-60 cells were obtained from the American Type Culture Collection (ATCC, Manassas, VA, USA). Human neonatal fibroblasts were obtained from LGC Standards (Luckenwalde, Germany). The HeLa and AGS lines were cultured in Dulbecco’s modified Eagle’s medium (DMEM) and DMEM/F12 medium, respectively. The HL-60 line was maintained in RPMI-1640 medium. Fibroblasts were cultured in Fibroblast Growth Medium with Supplement Mix. All the media were supplemented with 100 units/mL of penicillin, 100 µg/mL of streptomycin, and 10% (*v*/*v*) fetal bovine serum (FBS) (Merck Millipore, Burlington, MA, USA). The cells were incubated at 37 °C and 5% CO_2_.

#### 3.1.2. MTT Assay

To estimate the cytotoxic effect of the five compounds (**4**–**8**), MTT (3-(4,5-dimethylthiazol-2-yl)-2,5-diphenyltetrazolium bromide) assay was used [38]. Oxaliplatin was used as a positive control in this test. All the cell lines were seeded in 96-well plates at a density of 5 × 10^3^ cells/well and treated for 24 h with the compounds at concentrations of 0.5–40 µg/mL. Oxaliplatin was tested in the range of 20–400 µg/mL (50–1000 µM). After treatment, the cells were incubated with MTT (0.5 mg/mL; Merck Millipore, Burlington, MA, USA) for 3 h. Then, DMSO was used to dissolve formazan crystals, and the absorbance of the formazan solution was measured with a plate reader (Epoch, BioTek Instruments, Santa Clara, CA, USA). All the results (±standard deviation (SD)) were obtained from six repetitions in at least two independent experiments. The data are expressed as IC_50_ values (µg/mL).

#### 3.1.3. Annexin and Dead Cell Assay

To estimate the viability and induction of apoptosis/necrosis in the population of AGS cells treated with compound **5**, the Muse Annexin V and a Dead Cell Assay Kit (Merck Millipore, Burlington, MA, USA) were used [67]. The cells were seeded in 12-well plates (1 × 10^5^ cells/well) and incubated with the compound at concentrations of 1, 5, and 10 µg/mL. The concentration of DMSO added to the cells did not exceed 0.2% (*v*/*v*). After 24 h, the cells were stained with the kit reagents, namely, annexin V and 7-AAD (7-aminoactinomycin), and analyzed by flow cytometry (Muse Cell Analyzer, Merck Millipore, Burlington, MA, USA). The experiments were performed in three independent repeats.

#### 3.1.4. MitoPotential Assay

The AGS cells were seeded in a 12-well plate (1 × 10^5^ cells/well) and incubated with compound **5** at concentrations of 1, 5, and 10 µg/mL; oxaliplatin at the concentration of 40 µg/mL; and DMSO (0.2%, *v*/*v*). After 5 h of exposure, the cells were stained with the Muse MitoPotential Assay Kit (Merck Millipore, Burlington, MA, USA) and analyzed with the Muse Cell Analyzer (Merck Millipore, Burlington, MA, USA) [68]. The percentage of depolarized live and dead cells was estimated in the experiments, which were independently repeated three times.

#### 3.1.5. Cell Cycle Analysis

The AGS cells were seeded in a 6-well plate (5 × 10^5^ cells/well) and incubated with compound **5** at concentrations of 1, 5, and 10 µg/mL for 48 h; oxaliplatin at the concentration of 40 µg/mL; and DMSO (0.2%, *v*/*v*). After treatment, the cells were stained with propidium iodide from the Muse Cell Cycle Assay Kit (Merck Millipore, Burlington, MA, USA), and the amount of cells in each phase of the cell cycle was determined using the Muse Cell Analyzer (Merck Millipore, Burlington, MA, USA) [68]. The experiment was independently repeated three times.

#### 3.1.6. Caspase-8 and -9 Activity Assay

The AGS cells were seeded in a 96-well plate (5 × 10^3^ cells/well) and treated with compound **5** at concentrations of 1, 5, 7, and 10 µg/mL for 24 h; oxaliplatin at the concentration of 20 µg/mL; and DMSO (1%, *v*/*v*). The cells were then treated with reagents from Caspase-Glo 8 Assay or Caspase-Glo 9 Assay (Promega, Madison, WI, USA) and measured with the luminometer Glomax Multi + Detection System (Promega, Madison, WI, USA) [68]. The experiments were performed in two independent repeats in three repetitions.

#### 3.1.7. Statistical Analysis

Statistical analysis was performed with the STATISTICA 12.0 software package (Stat-Soft. Inc., Tulsa, OK, USA). All data are expressed as mean values ± standard deviation (±SD). The Student’s *t*-test was used to compare the results with the control sample. The statistical significance was set at *p* < 0.05.

### 3.2. Antioxidant Study

#### 3.2.1. Materials

Neutrophil elastase, ascorbic acid, N-succinyl-Ala-Ala-Ala-p-nitroanilide (SANA), oleanolic acid, ABTS (2,2-azino-bis(3-ethylbenzthiazoline-6-sulfonic acid)) diammonium salt, potassium persulfate, DPPH (2,2-diphenyl-1-picrylhydrazyl), and DMSO (dimethyl sulfoxide) were sourced from the Sigma Chemical Co. (St. Louis, MO, USA). TRIS-HCl (0.2 M, pH 8), HPLC-grade methanol, and chloroform were sourced from P.O.Ch (Gliwice, Poland).

#### 3.2.2. Elastase Assay

The elastase analysis was performed according to the spectrophotometric method [69,70] with oleanolic acid as a reference. The reaction mixture (Tris-HCl buffer (pH 8.0), 1.0 μg/mL porcine pancreatic elastase, and different concentrations of chalcone derivatives was preincubated at room temperature for 15 min. Afterwards, the addition of the substrate—SANA—started the reaction. Product formation changes were analyzed at λ = 410 nm for 20 min every 20 s (Epoch, BioTek Instruments, Santa Clara, CA, USA). The control was composed of Tris-HCl buffer (pH 8.0), an appropriate concentration of the chalcone derivative, and SANA.

#### 3.2.3. DPPH Assay

The DPPH radical scavenging assay of samples (ligands or complexes) was performed with ascorbic acid as a standard [38,69]. Different concentrations of chalcone derivatives, dissolved in DMSO, were mixed with the same amount of 0.06 mM DPPH methanolic solution and incubated at room temperature in the dark for 30 min. The absorbance was analyzed at λ = 517 nm using a 96-well microplate reader (Epoch, BioTek System, Winooski, VT, USA). The control was composed of DPPH and DMSO. DPPH inhibition was calculated according to the following Formula (2):DPPH Inhibition (%) = [(Acontrol − Asample)/Acontrol] × 100%(2)

#### 3.2.4. ABTS Assay

The ABTS radical scavenging assay of chalcone derivatives was performed with ascorbic acid as a standard [38,69]. Different concentrations of samples, dissolved in DMSO, were mixed with 170 μL of ABTS solution (2 mM ABTS diammonium salt and 3.5 mM potassium persulfate), and water was added to a final volume of 300 μL. ABTS solution with DMSO was used as a control. After 10 min of incubation at 30 °C in the dark, the absorbance changes were observed at λ = 750 nm with a 96-well microplate reader (Epoch, BioTek System, Santa Clara, CA, USA).

ABTS inhibition was calculated according to the following Formula (3):ABTS Inhibition (%) = [(Acontrol − Asample)/Acontrol] × 100%(3)

#### 3.2.5. β-Carotene Bleaching Test

β-Carotene bleaching test was based on the method described by Olszowy et al. [71] with some modifications. First, 1 mg of β-carotene was dissolved in 5 mL of chloroform. Afterwards, 25 mg/mL of linoleic acid and 200 mg of Tween 40 were added. The chloroform was evaporated, and 50 mL of oxygenized water was added. The created emulsion was mixed with different concentrations of chalcone derivatives. β-carotene bleaching was analyzed at λ = 492 nm before (t = 0 min) and after 90 min (t = 90 min) of incubation at 50 °C (Epoch, BioTek Instruments, Santa Clara, CA, USA). Ascorbic acid was used as a standard. The control was composed of emulsion and DMSO.

The results were calculated according to the following Formula (4):[(AsampleT0 − AsampleT90)/(AcontrolT0 − AcontrolT90) × 100%(4)

### 3.3. Antimicrobial Study

#### Methodology

The MIC determinations for *Staphylococcus epidermidis* ATCC14990, *Staphylococcus aureus* ATCC6538, *Enterococcus faecalis* ATCC51299, *Klebsiella pneumoniae* ATCC13883, *Escherichia coli* ATCC8739, *Salmonella enterica* ATCC13076, *Helicobacter pylori* ATCC43504, *Campylobacter jejuni* ZMF, *Campylobacter coli* ZMF, *Bacillus cereus* PCM 1948,2019 (ATCC11778), and *Listeria monocytogenes* PCM2191 were performed based on the methodology described by Krauze-Baranowska et al. [72]. The procedure for *H. pylori* was used in the experiments with *Campylobacter* strains, and the procedure for *B. subtilis* was used in the experiments with *L. monocytogenes* and *B. cereus*. The tested samples were dissolved in DMSO and diluted with water (concentration of approximately 500 μg/mL). The solutions were diluted geometrically by transferring the mixture/dilution (100 μL) from the first to the twelfth well of the microtiter plate. The final concentrations of samples used for antimicrobial effects ranged from 500 to 0.006 µg/mL. Ampicillin was used as a reference compound in the concentration range of 128 to 0.0625 µg/mL. The tests were incubated at 37 °C for 48 h. The diluent concentration had no effect on the activity of the tested compounds. All experiments were performed in triplicate.

### 3.4. Synthesis

The following instruments and parameters were used: melting points: Boetius PHMK apparatus; IR spectra: KBr pellets, 400–4000 cm^−1^ Thermo Mattson Satellite FTIR spectrophotometer; and ^1^H and ^13^C NMR: Varian Unity 500 plus apparatus at 500 MHz. Chemical shifts are expressed in parts per million (ppm) relative to TMS as an internal standard. Elemental analyses for C, H, and N were performed on a 2400 Series II CHN Elemental Analyzer (PerkinElmer, Shelton, CT, USA) and are in agreement with the theoretical values within ±0.4% range. Mass spectra were recorded on an LCMS 2010 spectrometer (Shimadzu, Japan). The compounds were identified based on their molecular ions obtained through electrospray negative ionization (ESI) (Appendix A). Thin-layer chromatography (TLC) was performed on Merck Kieselgel 60 _F254_ plates and visualized with UV. Gravity column chromatography was performed on Fluka silica gel 60 of particle size 35–75 µm and 220–440 mesh (Sigma-Aldrich Chemie, Steinheim, Germany). The starting compounds 4-aminoacetophenone (**1**) and 2,4-dichloro-5-methylbenzenesulfonyl chloride (**2**) are commercially available (Alfa Aesar, Haverhill, MA, USA).

The stability of chalcone derivative compounds was determined with a spectrophotometer (Epoch, BioTek Instruments, Santa Clara, CA, USA) in the range of 200–700 nm for 0–24 h. The compounds were dissolved in DMEM medium with 10% DMSO (*v*/*v*). An example UV–Vis spectrum of compound **5** is shown in the Appendix A.

#### 3.4.1. *N*-(4-Acetylphenyl)-2,4-Dichloro-5-Methylbenzenesulfonamide (**3**)

To 1.56 g (6 mmol) of 2,4-dichloro-5-methylbenzenesulfonyl chloride, 0.81 g (6 mmol) of 4-aminoacetophenone and 6 mL of pyridine were added. The substrates were stirred at a reflux temperature for 4 h. After this time, 30 mL of ice water was added to the reaction mixture. The resulting precipitate was filtered off, dried, and crystallized from the DMF/water mixture to produce pure product **3** in 92% yield as a beige solid: mp 237–238 °C; IR (KBr) ν_max_ 3199, 3091, 3059, 2946, 2879, 1672, 1599, 1362, 1165 cm^−1^; ^1^H NMR (500 MHz, DMSO-*d*_6_) δ: 2.37 (s, 3H, CH_3_), 2.45 (s, 3H, CH_3_), 7.17 (d, 2H, Ar, *J* = 8.8 Hz), 7.81 (s, 1H, Ar, H-3), 7.83 (d, 2H, Ar, *J* = 8.8 Hz), 8.15 (s, 1H, Ar, H-6), 11.26 (s, 1H, NHSO_2_) ppm; Anal. Calcd for C_15_H_13_Cl_2_NO_3_S (358.24): C, 50.29, H, 3.66, N, 3.91; found: C, 50.30, H, 3.68, N, 3.89.

#### 3.4.2. Synthesis (*E*)-2,4-Dichloro-*N*-(4-Cinnamoylphenyl)-5-Methylbenzenesulfonamide Derivatives **4**–**8**

A mixture of *N*-(4-acetylphenyl)-2,4-dichloro-5-methylbenzenesulfonamide (0.358 g, 1 mmol), potassium ethoxide (1.2 g KOH and 15 mL anhydrous ethanol), and the appropriate benzaldehyde derivative (1 mmol) was stirred at room temperature for 24–48 h. After this time, 10 mL of ice water was added to the reaction mixture and acidified with a dilute HCl solution of pH ~ 6. The resulting precipitate was filtered off and dried. The precipitated solid was obtained by filtration, and the products were recrystallized from toluene to give **4**–**8**.

##### (*E*)-2,4-Dichloro-*N*-(4-Cinnamoylphenyl)-5-Methylbenzenesulfonamide (**4**)

The compound was obtained as a light yellow solid in 90% yield: mp 216–217 °C; IR (KBr) ν_max_ 3173, 3021, 2922, 1646, 1604, 1351, 1163 cm^−1^; ^1^H NMR (500 MHz, DMSO-*d*_6_) δ: 2.31 (s, 3H, CH_3_), 6.79 (d, 2H, Ar, *J* = 8.8 Hz), 7.38–7.43 (m, 3H, Ar), 7.48 (s, 1H, Ar, H-3), 7.56 (d, 1H, HC=CH, *J* = 15.6 Hz), 7.79 (m, 4H, Ar, *J* = 8.8 Hz), 7.83 (d, 1H, CH=CO, *J* = 15.6 Hz), 7.95 (s, 1H, Ar, H-6), 11.25 (s, 1H, NHSO_2_) ppm; ^13^C NMR (DMSO-*d*_6_, 125 Hz) δ: 19.54, 117.86, 122.26, 129.22, 129.26, 129.34, 130.75, 131.00, 132.05, 133.00, 134.04, 135.14, 135.19, 136.56, 139.68, 141.86, 143.97, 187.93 ppm; Anal. Calcd C_22_H_17_Cl_2_NO_3_S (445.03): C, 59.20, H, 3.84, N, 3.14; found: C, 59.97, H, 3.88, N, 3.19. MS (ESI) *m*/*z*: 444 [M-1]^−^.

##### (*E*)-*N*-{4-[3-(4-Bromophenyl)acryloyl]phenyl}-2,4-Dichloro-5-Methylbenzenesulfonamide (**5**)

The compound was obtained as a light yellow solid in 58% yield: mp 240–242 °C; IR (KBr) ν_max_ 3183, 3079, 2925, 1651, 1604, 1351, 1160; ^1^H NMR (500 MHz, DMSO-*d*_6_) δ: 2.49 (s, 3H, CH_3_), 7.22 (d, 2H, Ar, *J* = 8.8 Hz), 7.62–7.66 (m, 3H, Ar), 7.80–7.82 (m, 3H, Ar), 7.89 (d, 1H, CH=CO, *J* = 15.6 Hz), 8.05 (d, 2H, Ar, *J* = 8.8 Hz), 8.18 (s, 1H, Ar, H-6), 11.33 (s, 1H, NHSO_2_) ppm; ^13^C NMR (DMSO-*d*_6_, 125 Hz) δ: 19.55, 117.85, 123.07, 124.34, 129.23, 130.79, 131.16, 132.04, 132.30, 132.81, 134.02, 134.46, 135.26, 136.54, 139.63, 142.10, 142.55, 187.80 ppm; Anal. Calcd C_22_H_16_BrCl_2_NO_3_S (522.94): C, 50.31, H, 3.07, N, 2.67; found: C, 50.37, H, 3.00, N, 2.62. MS (ESI) *m*/*z*: 522 [M-1]^−^.

##### (*E*)-2,4-Dichloro-*N*-{4-[3-(4-Fluorophenyl)acryloyl]phenyl}-5-Methylbenzenesulfonamide (**6**)

The compound was obtained as a white solid in 87% yield: mp 225–226 °C; IR (KBr) ν_max_ 3194, 3083, 2923, 1652, 1608, 1353, 1154 cm^−1^; ^1^H NMR (500 MHz, DMSO-*d*_6_) δ: 2.39 (s, 3H, CH_3_), 7.22 (d, 2H, Ar, *J* = 8.8 Hz), 7.27 (t, 2H, Ar, *J* = 8.8 Hz), 7.67 (d, 1H, HC=CH, *J* = 16.0 Hz), 7.81 (s, 1H, Ar, H-3), 7.82 (d, 1H, CHCO, *J* = 16.0 Hz), 7.91–7.94 (m, 2H, Ar), 8.05 (d, 2H, Ar, *J* = 8.8 Hz), 8.18 (s, 1H, Ar, H-6), 11.32 (s, 1H, NHSO_2_) ppm; ^13^C NMR (DMSO-*d*_6_, 125 Hz) δ: 19.55, 116.29, 116.46, 117.86, 122.19, 129.23, 130.76, 131.59, 131.66, 131.84, 131.87, 132.06, 132.97, 134.04, 135.22, 136.57, 139.68, 141.92, 142.73, 162.83, 164.81, 187.85 ppm; Anal. Calcd C_22_H_16_Cl_2_FNO_3_S (463.02): C, 56.91, H, 3.47, N, 3.02; found: C, 56.87, H, 3.38, N, 3.02. MS (ESI) *m*/*z*: 462 [M-1]^−^.

##### (*E*)-2,4-Dichloro-*N*-{4-[3-(4-Chlorophenyl)acryloyl]phenyl}-5-Methylbenzenesulfonamide (**7**)

The compound was obtained as a light yellow solid in 55% yield: mp 249–250 °C; IR (KBr) ν_max_ 3187, 3079, 2926, 1651, 1605, 1351, 1159 cm^−1^; ^1^H NMR (500 MHz, DMSO-*d*_6_) δ: 2.39 (s, 3H, CH_3_), 7.23 (d, 2H, Ar, *J* = 8.8 Hz), 7.50 (d, 2H, Ar_._, *J* = 8.3 Hz), 7.66 (d, 1H, CH=CHCO, *J* = 15.6), 7.80 (s, 1H, Ar, H-3), 7.86–7.89 (m, 3H, Ar), 8.06 (d, 2H, Ar, *J* = 8.3 Hz), 8.18 (s, 1H, Ar, H-6), 11,32 (s, 1H, NHSO_2_) ppm; ^13^C NMR (DMSO-*d*_6_, 125 Hz) δ: 19.54, 117.83, 123.02, 129.23, 129.34, 129.37, 130.79, 130.95, 132.04, 132.88, 134.04, 134.14, 135.19, 135.46, 136.56, 139.68, 141.96, 142.47, 187.81 ppm; Anal. Calcd C_22_H_16_Cl_3_NO_3_S (478.99): C, 54.96, H, 3.35, N, 2.91; found: C, 54.90, H, 3.38, N, 2.89. MS (ESI) *m*/*z*: 478 [M-1]^−^.

##### (*E*)-2,4-Dichloro-5-Methyl-*N*-{4-[3-(*p*-Tolyl)acryloyl]phenyl}benzenesulfonamide (**8**)

The compound was obtained as a light yellow solid in 77% yield: mp 229–230 °C; IR (KBr) ν_max_ 3186, 3088, 2980, 2944, 1654, 1599, 1353, 1162 cm^−1^; ^1^H NMR (500 MHz, DMSO-*d*_6_) δ: 2.34 (s, 3H, CH_3_), 2.39 (s, 3H, CH_3_), 7.21–7.26 (m, 4H, Ar, *J* = 8.8 Hz), 7.64 (d, 1H, HC=CH, *J* = 16.0 Hz), 7.72 (d, 2H, Ar, *J* = 8.8 Hz), 7.79 (d, 1H, CHCO, *J* = 15.6 Hz), 7.81 (s, 1H, Ar, H-3), 8.04 (d, 2H, Ar, *J* = 8.8 Hz), 8.17 (s, 1H, Ar_._, H-6), 11.27 (s, 1H, NHSO_2_) ppm; ^13^C NMR (DMSO-*d*_6_, 125 Hz) δ: 19.55, 21.52, 118.06, 121.25, 128.65, 129.25, 129.27, 129.34, 129.96, 130.64, 131.95, 132.47, 133.89, 136.36, 139.25, 140.99, 143.83, 187.76 ppm; Anal. Calcd C_23_H_19_Cl_2_NO_3_S (459.05): C, 60.00, H, 4.16, N, 3.04; found: C, 60.41, H, 4.06, N, 3.09. MS (ESI) *m*/*z*: 458 [M-1]^−^.

## 4. Conclusions

We developed a method for the synthesis of a novel series of (*E*)-2,4-dichloro-*N*-(4-cinnamoylphenyl)-5-methylbenzenesulfonamide derivatives containing the following two pharmacophore groups: chalcone and benzenesulfonamide. We evaluated their anticancer, antioxidant, and antimicrobial activities. Initially, the anticancer efficacy of the obtained compounds **4**–**8** was tested against the following human cells: HeLa, HL-60, AGS, and fibroblasts. Among cancer cell lines, compound **5** was highly potent and exhibited higher cytotoxicity on cancer cells than noncancerous fibroblasts. This chalcone derivative was chosen for further experiments to determine cell cycle inhibition and the type of cell death induced in AGS cells. The results showed that derivative **5** significantly inhibited the cell cycle and triggered cell death through depolarization of the mitochondrial membrane and activation of caspase-8 and -9. These findings indicate that compound **5** induces apoptosis in gastric cancer cells by both the extrinsic and intrinsic pathways. Also, in this study, we estimated whether the tested compounds would be capable of participating in free radical reactions and therefore protecting macromolecules and lipid membranes from oxidative stress. In these tests, the strongest antioxidant properties were demonstrated by compound **5**. In the neutrophil elastase inhibition assay, the best activity, comparable to the effect of oleanolic acid, was demonstrated by compounds **7** and **8**. These results indicated their significant anti-inflammatory potential; however, this should be developed in future studies.

Analysis of the structure–activity relationship of 2,4-dichloro-*N*-(4-cinnamoylphenyl)-5-methylbenzenesulfonamide indicated that the presence of bromine in the R position contributed to the strongest anticancer effect. Moreover, the alkenyl moiety played a crucial role in the antioxidant activity of the hybrids.

To determine the antibacterial activity of derivatives **4**–**8**, MIC values were estimated, and they indicated no significant effect on the tested bacterial strains.

In summary, our results indicate that the effect of chalcone–sulfonamide hybrids **4**–**8** on cancer cells should be further developed, especially in the case of compound **5**. In-depth studies are required to identify specific genes at the mRNA level and proteins involved in the cell death process.

## Data Availability

Data are contained within the article and Appendix A.

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
