# Peer review of "New Chalcone Derivatives Containing 2,4-Dichlorobenzenesulfonamide Moiety with Anticancer and Antioxidant Properties"

_ijms, 2023, doi:10.3390/ijms25010274_

Round 1

Reviewer 1 Report

Comments and Suggestions for Authors

The article is devoted to the synthesis of new derivatives of chalcones with sulfonamides. The work is a continuation of a series of studies by a group of authors on the preparation of chalcone derivatives and investigation of their biological properties. The article is well structured and designed. The obtained derivatives are characterized by spectral methods. We would especially like to note a detailed study of the biological activity of the obtained substances. The work will certainly be of interest to researchers due to its relevance and a large amount of experimental data. 

"New chalcone derivatives containing 2,4-dichlorobenzenesulfonamide moiety with anticancer and antioxidant properties" by Anita Bułakowska, Jarosław Sławiński, Anna Hering, Magdalena Gucwa, J. Renata Ochocka, Rafał Hałasa, Justyna Stefanowicz-Hajduk. 

This work considers the possibility of obtaining and detailed study of biological activity (including anticancer activity) of new chalcone derivatives with sulfonamides, namely (E)-2,4-dichloro-N-(4-cinnamoylphenyl)-5-methylbenzenesulfonamides containing two pharmacophore groups: 2,4-dichlorobenzenesulfonamide and chalcone. The authors evaluated the antitumor effect of the obtained samples against three types of cancer cells and carried out experiments to identify possible mechanisms of their anticancer action. The work is a continuation of a series of studies by a group of authors on the preparation of chalcone derivatives and investigation of their biological properties. The relevance of this work is due to the prospects of using chalcone derivatives and sulfonamides as cytostatic drugs and the search for new candidate molecules for these purposes.

This article describes the method of preparation of chalcone derivatives with sulfonamides, evaluates the structure of the obtained compounds and determines their pronounced cytotoxic effect against common cancer cell lines, which allows us to consider the obtained compounds as potential agents for combating cancer.  The article is well structured and designed. The obtained derivatives are characterized by spectral methods. The conclusions presented by the authors are in line with the main results and objectives of the paper. All references are appropriate. There is no excessive self-citation.  The following suggestions can be mentioned as possible suggestions for the improvement of the manuscript:

1) Considering the low values of antiradical activity of the obtained compounds, it is possible either to remove the obtained results from the discussion or to explain such low values (the initial substances - flavonoids have a pronounced antiradical activity).

2) It is necessary to revise the figures and make their design more uniform, increase the font, and contrast, especially Figure 2.

3) Considering the high level of the IJMS journal, it would be good to accompany the description of the obtained samples of derivatives with the data of X-ray diffraction analysis, which allows to reliably establish the structure of the obtained compounds.

4) In the introduction, the authors should provide more details specifically on the consideration of more examples of successful preparation of chalcone derivatives with sulfonamides. 

Author Response

Dear Editor,

We would like to thank for critical reading this manuscript and valuable suggestions. We have carefully considered all of the suggestions and made the appropriate additions. The changes are marked in red in the text.

Reviewer 1 

Comment 1: Considering the low values of antiradical activity of the obtained compounds, it is possible either to remove the obtained results from the discussion or to explain such low values (the initial substances - flavonoids have a pronounced antiradical activity).

Response: We have explained the low antiradical activity of compounds in more detail (lines 427-435).

Comment 2: It is necessary to revise the figures and make their design more uniform, increase the font, and contrast, especially Figure 2.

Response: We have improved figures as much as possible, increased the font in graphs presented the results as average values. We show the cells in Figure 2 under bigger magnification (200x). In the case of graphs copied directly from flow cytometry software (Fig. 3-5), unfortunately it is not possible to change the font size.

Comment 3: Considering the high level of the IJMS journal, it would be good to accompany the description of the obtained samples of derivatives with the data of X-ray diffraction analysis, which allows to reliably establish the structure of the obtained compounds.

Response: The authors agree with the reviewer that it would be useful to solve the crystal structure, provided that a crystal of the right size could be obtained, which is not certain. Obtaining the appropriate crystal size and conducting this type of research will certainly require more than 10 days. Therefore, we decided to perform MS spectra. Descriptions of the spectra are provided in section 3.4. Synthesis. The spectra are included in Supplementary Materials. 

Comment 4: In the introduction, the authors should provide more details specifically on the consideration of more examples of successful preparation of chalcone derivatives with sulfonamides.

Response: As suggested by the reviewer, more examples of successful preparation of chalcone derivatives with sulfonamides are included in the introduction (lines 83-101).

Reviewer 2 Report

Comments and Suggestions for Authors

The present study describes the synthesis and biological use of new compounds derived from chalcone-sulfonamide structure. The authors aimed to estimate the potential use of newly synthesized derivatives  as anticancer drugs. Thus, they estimated the biological properties of the compounds using several biochemical tests assessing cell proliferation, anticancer effect, antioxidant properties, proapoptotic acting etc. Although the study can be considered as potentially important, some limitations should be discussed and additional results might be included into the manuscript.

Major comments:

1) If the compound "5" should be considered for antitumour use, the mechanism of its cellular effect should be described in more detail to provide the readers with additional information about the principle of its acting.

2) What is the stability of synthesized compounds in hydrophilic environment? Is there any hydrolysis occurring, or any other reaction changing the structure of tested compounds?

3) The biological/biochemical tests have been performed at different times of incubation with tested compounds, i.e. MTT 24 hours, mitochondrial depolarization 5 h, cell cycle arrest 48 h, etc. What is the reason? Which results were obtained using MTT after 48 h of incubation? It would be more appropriate to provide the results at similar time periods and, in addition, to provide some others to describe the biological effects in more detail at additional durations of incubation.

4) What would be the biological activity of parent compounds, i.e. an appropriate chalcone and a substituted sulfonamide? Is there any benefit to use structurally bound chalcone-sulfonamide derivative in comparison to the use of chalcone/sulfonamide without structural binding?

Minor comments:

1) English could be improved in some parts of the text - e.g. chapter 2.2.2 - "DPPH and ABTS assays are simple colorimetric method,..." etc.

2) the authors determined the chalcones as "polyphenols" (Introduction line 47) - is that proper meaning regarding the structure of tested compounds?

3) the legends of illustration should be included on the same page as the illustration

4) Fig 2 - the magnification of the photomicrographs should be increased to show the cells in more detail

5) testing of antimicrobial activity - the authors present only negative results, thus, the Table 4 could be deleted or added to the Supp. file, especially, if the topical aim of the study was to test antitumour activity.

6) What was the reason to use neonatal fibroblasts instead of fibroblasts from an adult?

Comments on the Quality of English Language

noted above

Author Response

Dear Editor,

We would like to thank for critical reading this manuscript and valuable suggestions. We have carefully considered all of the suggestions and made the appropriate additions. The changes are marked in red in the text.

Reviewer 3

Comment 1: If the compound "5" should be considered for antitumour use, the mechanism of its cellular effect should be described in more detail to provide the readers with additional information about the principle of its acting.

Response: We have described the mechanism of action of compound 5 in details (lines 314-327). We have also estimated the activation of caspase-9 in AGS cells treated with compound 5 to confirm the involvement of mitochondrial apoptotic pathway in cell death. The results are shown in Figure 6 (caspase-8 and -9). 

Comment 2: What is the stability of synthesized compounds in hydrophilic environment? Is there any hydrolysis occurring, or any other reaction changing the structure of tested compounds?

Response: The stability of synthesized compounds we have estimated based on UV-Vis spectra done at different times. We have shown an example spectrum for compound 5 attached in the Supplementary Materials. We did not observe changes in the structures of chalcone derivatives in hydrophilic environment during experiments.

Comment 3: The biological/biochemical tests have been performed at different times of incubation with tested compounds, i.e. MTT 24 hours, mitochondrial depolarization 5 h, cell cycle arrest 48 h, etc. What is the reason? Which results were obtained using MTT after 48 h of incubation? It would be more appropriate to provide the results at similar time periods and, in addition, to provide some others to describe the biological effects in more detail at additional durations of incubation.

Response: Thank you for this comment. Generally, in such experiments the time of incubation may be different for the reason that particular cellular factors usually change during cell death at different times. Thus, we showed their activity when it was significant and important to explain the mechanisms of action of compound 5. In the case of mitochondrial membrane potential assay, the main aim of this test is to show live cells with depolarized mitochondria, not dead. After 24 h, we obtained too much dead cells and a little live cells with depolarized mitochondria. Thus, 24 h as the time of incubation was too long for this experiment to show the results. According to MTT, this test was performed at incubation time where, if compounds are cytotoxic, any changes should be observed. For the cell cycle test, it is recommended to show the results even after about 48 h, which is needed for completion of the entire cell cycle, thus the incubation time in this kind of experiment should be even up to 48 h. We normally carry out this assay as well as MTT test at these incubation times. Also, a day of incubation was enough to show participation of caspase-8 and -9 in cell death of AGS. This activation was significant compared to the untreated cells, so the time 48 h would be too long in such an experiment.  

Comment 4: What would be the biological activity of parent compounds, i.e. an appropriate chalcone and a substituted sulfonamide? Is there any benefit to use structurally bound chalcone-sulfonamide derivative in comparison to the use of chalcone/sulfonamide without structural binding?

Response: We did not evaluate the biological activity of the parent compounds, i.e. the corresponding chalcone and substituted sulfonamide. We were basing on literature reports which proved that the combination of sulfonamide and chalcone scaffolds in compounds resulted in increased biological activity of sulfonamidochalcones compared to chalcones themselves. We adopted the "one drug, many targets" concept by synthesizing such compounds. Each of these pharmacophores possesses characteristic chemical reactivities and a set of biological properties that can serve as multifunctional drugs or multi-targeted ligands. 

Comment 5: English could be improved in some parts of the text - e.g. chapter 2.2.2 - "DPPH and ABTS assays are simple colorimetric method,..." etc.

Response: Thank you for this comment. The manuscript has been linguistically proofread by a native speaker. We have attached the certificate confirming this revision.

Comment 6: the authors determined the chalcones as "polyphenols" (Introduction line 47) - is that proper meaning regarding the structure of tested compounds?

Response: According to the definition of chalcones, they are a group of plant-derived polyphenolic compounds belonging to the flavonoids family (https://www.sciencedirect.com/topics/chemistry/chalcone).

Comment 7: the legends of illustration should be included on the same page as the illustration.

Response: We have corrected this.

Comment 8: Fig 2 - the magnification of the photomicrographs should be increased to show the cells in more detail.

Response: We have corrected this Figure and showed the cells under bigger magnification (x200).

Comment 9: testing of antimicrobial activity - the authors present only negative results, thus, the Table 4 could be deleted or added to the Supp. file, especially, if the topical aim of the study was to test antitumour activity.

Response: We have added this table to the Supplementary Materials.

Comment 10: What was the reason to use neonatal fibroblasts instead of fibroblasts from an adult?

Response: We used neonatal fibroblasts since we have this kind of non-cancerous cell line in our biobank. There were no significant reason to use this cell line, only to show the effect of the tested compounds on normal cells.

Thank you for your time and consideration.

Dr. Anita Bułakowska